# NAS Without Priors: A Robust Architecture Search Framework for Unseen-Data

**Shahid Siddiqui**                                                                 *msiddi01@ucy.ac.cy*
*Electrical and Computer Engineering Department, University of Cyprus*
*Thaka Advanced Information Technology, Saudi Arabia*

**Christos Kyrkou**                                                           *kyrkou.christos@ucy.ac.cy*
*KIOS Research and Innovation Center of Excellence, University of Cyprus*

**Theocharis Theocharides**                                                  *ttheocharides@ucy.ac.cy*
*Electrical and Computer Engineering Department, University of Cyprus*
*KIOS Research and Innovation Center of Excellence, University of Cyprus*

**Reviewed on OpenReview:** *https://openreview.net/forum?id=PaEk1gYrFz*

## Abstract

Neural architecture search (NAS) has been widely used to automate neural network design for image classification; however, most NAS research has focused on CIFAR, ImageNet, and their derivative benchmarks. These datasets benefit from well-established architecture design practices, data preprocessing techniques, and training protocols developed prior to NAS, causing many NAS methods to meta-overfit and struggle to generalize to entirely novel datasets. In this work, we analyze the limitations of existing NAS practices and propose a framework specifically designed to generalize to unseen-data. In contrast to the prevailing paradigm of exploring extremely large search spaces using low-fidelity evaluations, we advocate sparser exploration combined with high-fidelity performance estimation. We demonstrate that macro-architecture variations alone induce substantial architectural diversity, and that concentrating computational resources on high-fidelity evaluation of fewer candidates produces reliable reward signals enabling better architecture discovery. To obtain robust candidate rankings, we repeatedly train architectures on the entire training set using multiple random seeds. While this approach substantially reduces performance variance due to random seed variability and enables accurate candidate ranking, it comes at a significant computational cost. To mitigate the cost of such high-fidelity evaluation, particularly for larger or high-resolution datasets, we introduce a dataset- and architecture-aware multi-fidelity search strategy that both reduces computational overhead and stabilizes candidate rankings under varying fidelity levels. We evaluate our framework on the NAS Unseen-Data Challenge[1], where, under an 8-hour budget per dataset (including search and training), it ranks first with a combined score of 12.19, outperforming both manually designed baselines and the second- and third-place NAS solutions (10.89 and 10.43). Moreover, we test our framework on CIFAR-10 to enable comparison with the broader NAS literature, despite our primary focus on NAS generalization beyond conventional benchmarks. Our framework achieves competitive error rates against a broad spectrum of macro- and micro-NAS methods, demonstrating the effectiveness of sparse, high-fidelity evaluations and the proposed multi-fidelity search algorithm. Our code and search logs are available at: https://github.com/siddikui/NAS-Challenge-2025-Solution.

---

[1]NAS Unseen-Data Challenge 2025 Leaderboard

# 1 Introduction

Neural architecture search (NAS) aims to automatically discover an optimal neural network architecture for a given dataset. Existing NAS works (Zoph & Le, 2016; Baker et al., 2016; Real et al., 2017; Suganuma et al., 2017; Zoph et al., 2018; Liu et al., 2018) have produced competitive results on CIFAR (Krizhevsky et al., 2009) and ImageNet (Russakovsky et al., 2015) datasets; however, prior advances in data augmentation and training tricks have been shown to contribute more substantially to accuracy gains than the NAS methods themselves (Geada et al., 2024; Yang et al., 2019). Moreover, even NAS-specific benchmarks (Ying et al., 2019; Dong & Yang, 2020) are derived from these same datasets, and continued reliance on them has limited the applicability of NAS to truly novel tasks (Mehta et al., 2022). For instance, the widely adopted cell-only search paradigm (Zoph et al., 2018; Liu et al., 2018) does not generalize well to newer datasets, as macro-architecture must still be manually specified. While effective macro-level configurations for CIFAR and ImageNet were established through prior expert knowledge, the optimal settings for a new dataset are generally unknown. Further, despite encompassing an enormous number of candidate architectures (e.g., $10^{35}$ and $10^{18}$ in Zoph et al. (2018) and Liu et al. (2018), respectively), such search spaces are known to exhibit limited performance variance (Yang et al., 2019). In contrast, early NAS works (Zoph & Le, 2016; Baker et al., 2016; Real et al., 2017; Cai et al., 2018) enable exploration of the entire architecture, but navigating these large spaces using inefficient evaluation schemes renders them computationally impractical (e.g., Zoph & Le (2016) trained 12,800 architectures for 50 epochs each, requiring 22400 GPU days). Table 1 highlights the substantial differences between conventional NAS benchmarks and unseen datasets introduced in the latest NAS Unseen-Data challenge (Geada et al., 2024; Towers et al., 2025a), indicating that widely used NAS assumptions and design choices are not readily applicable to unseen data.

The extreme computational cost of NAS, driven by enormous search spaces, has motivated the development of numerous techniques for accelerating candidate evaluation. One simple strategy is learning-curve extrapolation, which evaluates architectures using only a few training epochs (Domhan et al., 2015; Yan et al., 2021; Baker et al., 2017); however, newer datasets can exhibit substantially different convergence behaviors, making it difficult to predefine a fixed early stopping criterion that generalizes reliably. Another widely used low-fidelity approximation trains candidates on a subset of the data (Klein et al., 2017; Prasad et al., 2024; Shim et al., 2021; Na et al., 2021), but identifying a representative subset, one that yields performance comparable to full-dataset training, implicitly requires knowledge of the highest achievable accuracy, which is unknown *a priori* for unseen datasets. Similarly, model-based performance predictors (Wen et al., 2020; White et al., 2021b) rely on architecture–accuracy pairs that must be collected anew for each dataset, limiting their practicality for newer datasets. More aggressive speedup strategies attempt to estimate architecture performance with minimal or no training (Krishnakumar et al., 2022); however, decoupling evaluation from the target dataset undermines the core objective of NAS, that is, discovering dataset-specific architectures. Other approaches transfer weights between architecturally similar candidates (Real et al., 2017; Chen et al., 2016), introducing bias by favoring candidates evaluated later in the search, while supernet based methods (Saxena & Verbeek, 2016; Liu et al., 2018) rely on weight sharing that has been shown to produce inaccurate candidate rankings (Yang et al., 2019; Yu et al., 2019). Overall, existing evaluation schemes largely prioritize accelerating the assessment of large candidate sets, at the expense of accurate relative performance estimates of competing architectures and do not generalize well to unseen datasets, as summarized in Table 1.

In this work, we revisit existing NAS search spaces and evaluation strategies, arguing that allocating computational resources to obtain reliable rankings over a smaller set of candidates is more effective than conducting a large number of noisy evaluations, particularly when large candidate sets exhibit limited performance variance. Accordingly, we first analyze why existing search spaces do not generalize well and propose a substantially smaller yet dataset-adaptive design space with higher performance variability. We then examine candidate evaluation and address a well-known but largely overlooked issue of high performance variability across different random seeds (Li & Talwalkar, 2020; Yang et al., 2019; Lindauer & Hutter, 2020). Prior work has mainly treated this variability as a reproducibility concern, leading to recommendations for reporting averaged results across seeds. In contrast, we emphasize its impact during the search process itself, where reliance on a single seed can directly bias architecture selection. Ideally, to some extent, architecture discovery should be dataset-responsive rather than seed-sensitive. To achieve reliable candidate ranking, we incorporate multi-seed evaluation directly into the search process, evaluating each candidate across multiple

Table 1: Comparison of NAS search spaces, search strategies and evaluation schemes for standard seen versus unseen data introduced in the 5[th] NAS Unseen-Data Challenge 2025 Leaderboard.

| Dataset | Name-Type Dims-Size | Search Spaces | Evaluation Strategies | Search Strategies |
|---|---|---|---|---|
| **Seen** | CIFAR-Objects 3x32x32-10 50K/10K ImageNet-Objects 3x224x224-1K 12M/100K | 1. Insights into data characteristics e.g. 'color images with spatially connected shape features' influence search space design from state-of-the-art hand-crafted architectures for CIFAR10 and ImageNet. 2. Known data dimensions and effective parameters/FLOP budget allows fixing network outer skeleton such as stem, reduction blocks and terminal layers. This also allows constructing supernet search spaces. | 1. Known number of train-val samples, dimensions, evaluation cost, and expected validation performance allows low-fidelity schemes e.g. using 50% data with 25-50 epochs for CIFAR10. 2. Established augmentations such as random crop and horizontal flip and training settings such as epochs, batch size, learning rate and scheduling, momentum, weight decay, and cutout. | 1. Differentialble NAS suffers from stability issues e.g. inaccurate relative rankings. 2. Blackbox strategies such as RL, and evolution work well assuming all candidates comparable under a single evaluation protocol. 3. Joint architecture and training hyperparameter search (JAHS) is desirable however the search becomes expensive. |
| **Unseen** | Conway-Synthetic 1x10x10-25 50K/10K Cryptic-Language 1x6x768-2 59K/17K Windspeed-Weather 12x32x32-20 20K/7K | 1. The underlying nature of patterns is unknown and varies e.g. spatial clues are relevant for Conway but not for Windspeed. Prior state-of-the-art baseline networks are unavailable. 2. Non-trivial data dimensions and output classes demand flexibility in outer skeleton. Required params/FLOPS for a given dataset are also unknown. Since no prior information about macro-architecture, a generalizable search super-network is unavailable. Hence macro search spaces are more suitable. | 1. Unknown dimensions, and train-val sizes, incur substantially varying eval costs across datasets. Unknown convergence behaviors hinder specifying a generalized termination criteria. Random data subsets may not be representative if fewer or unbalanced samples per class. 2. CIFAR/ImageNet augmentations do not generalize to unseen data and unseen data may Over-/under- fit under existing protocols. | 1. One-shot NAS such as OFA is possible but initial supernet training is expensive for larger dimension datasets and the subnetworks may still require evaluation to find the optimal one. 2. Varying datasets with unknown training protocols require adaptive eval strategies. 3. Multi-fidelity search is applicable allowing evaluation at various training fidelities. This is similar to JAHS but implicit and partial hyperparameter search. |

seeds and ranking architectures based on their best observed performance. Furthermore, unlike existing approaches, we leverage the entire training set during search, hence, reducing the data-gap between search and final training, and ensuring correct suitability of the discovered architecture for the given dataset. While this high-fidelity evaluation improves robustness, it remains computationally expensive even for a small number of candidates; therefore, we introduce a multi-fidelity search algorithm that reduces overall search cost while further ensuring better architecture selection among competing architectures.

Multi-fidelity search algorithms, such as Successive Halving (SH) (Jamieson & Talwalkar, 2016), Hyperband (HB) (Li et al., 2018), and Bayesian Optimization with Hyperband (BOHB) (Falkner et al., 2018), progressively increase the evaluation budget for promising candidates, unlike majority of fixed-budget NAS works (Zoph & Le, 2016; Real et al., 2017; Liu et al., 2018; Domhan et al., 2015; Klein et al., 2017). Although originally designed to speed up the search, some studies have shown that jointly searching for architecture and training protocol yields better results (Yang et al., 2019; Bansal et al., 2022; Zela et al., 2018). For unseen data, where training protocols are also unknown, multi-fidelity evaluation can serve dual purpose; speeding up evaluation as well as allocating multiple training protocols to candidates for more reliable assessments. However, existing multi-fidelity methods typically allocate increased budgets only to high-performing candidates, potentially disadvantaging models that may require longer training to reach their full potential. Moreover, they either rely on unguided random search (Jamieson & Talwalkar, 2016; Li et al., 2018) or build surrogate performance models (Falkner et al., 2018), which can become costly for larger datasets. To address these limitations, we propose a novel search strategy that allocates dataset- and model-adaptive fidelity budgets. The strategy consists of two stages. In Phase 1, we collect a pool of promising candidates for a given dataset using local search, which has been shown to be a fast and competitive baseline for NAS (Den Ottelander et al., 2021; White et al., 2021a). These candidates are then further evaluated in Phase 2 using a model-aware fidelity allocation scheme, where the best architecture is identified through iterative elimination. The proposed strategy enables the discovery of dataset-specific architectures by ensuring fair comparison across models of varying sizes while remaining computationally feasible.

Our contributions can be summarized as follows:

- We revisit common NAS practices that rely on low-fidelity performance estimation over large search spaces and examine their limitations when applied to unseen datasets, motivating the use of higher-fidelity evaluation with a smaller set of candidates.

- We show that competitive, dataset-adaptive architectures can be discovered using only two macro-architectural variables: network depth and width.

- We employ a high-fidelity evaluation strategy that uses multi-seed evaluation and full training set to obtain robust candidate rankings.

- We introduce a dataset- and architecture-aware multi-fidelity search algorithm that reduces the cost of high-fidelity evaluation while maintaining reliable comparisons across candidates.

To evaluate our framework, we follow the NAS Unseen-Data Challenge (Geada et al., 2024; Towers et al., 2025a), which was introduced to assess NAS generalization beyond standard benchmarks. Participants are required to design a complete NAS pipeline and apply it to three entirely new and previously unseen datasets. Our proposed approach outperforms the manually designed baselines on all three datasets and achieves the highest combined score of 12.19, surpassing the second- and third-place solutions (10.89 and 10.43, respectively). Moreover, to facilitate comparison with the broader NAS literature, we conduct extensive experiments on CIFAR-10. Our framework outperforms several macro- and micro-NAS methods, highlighting the advantages of macro-architecture search over micro-architecture search. Further, within the DARTS search space, our approach achieves competitive performance, validating the effectiveness of the proposed multi-fidelity search strategy under a widely adopted NAS benchmark.

## 2 Related Work

The **search space** defines the set of neural architectures that the NAS method is allowed to explore, while the **search strategy** determines how candidate architectures are selected and refined within this space. To identify architectures that maximize performance on a given dataset, **candidate evaluation** scheme estimates their performance and provides a reward signal to search strategy. In the following, we review related work corresponding to each of these components.

### 2.1 Search Space

We categorize existing search spaces into three types: **macro-micro**, **cell-based** and **macro-only** and observe pros and cons associated with each. For a similar, yet detailed categorization of search spaces, we refer the reader to White et al. (2023). From here on, a search space may simply be referred to as space.

We refer to macro-micro spaces as those which allow searching for networks' macro (e.g. depth, width, skip connections) as well as micro/layer-level architecture (e.g. kernel size, layer type, strides). We list some of the prominent relevant spaces in Table 2. Some of these works are inspired by ResNet architecture (He et al., 2016) and incorporate skip connections resulting in multi-branch candidates (Zoph & Le, 2016; Real et al., 2017; Elsken et al., 2017; Kandasamy et al., 2018), while the rest follow chain-structured architecture paradigm where a network is simply composed of a sequence of layers. Moreover, most spaces incorporate kernel size with some works additionally searching for strides, pooling and fully-connected layers (Baker et al., 2016; Cai et al., 2018; Kandasamy et al., 2018).

To enhance search efficiency, Zoph et al. (2018), Zhong et al. (2018), and Liu et al. (2018) restrict the search to discovering transferable network building blocks (cells), while macro-architecture i.e. depth, width, and reduction blocks are chosen manually. While discovered cells transfer well across datasets, the corresponding macro-architectures vary significantly with dataset characteristics. For instance, Liu et al. (2018) manually choose 20 layers, 36 initial channels and two reduction blocks for CIFAR-10, but switches to 14 layers, 48 initial channels, and 4 reduction blocks for ImageNet. *This illustrates that as dataset difficulty changes, macro-architecture become increasingly critical, whereas the cell remains transferable.* Duan et al. (2021) jointly search for cells and macro-architecture to enhance NAS generalizability to diverse tasks. However, Wan et al. (2022); Yang et al. (2019) have shown that cell-only search spaces tend to exhibit limited performance variance despite containing an enormous number of candidates (e.g., on the order of $10^{18}$ architectures in Liu et al. (2018)), indicating that most of the performance variance is a result of macro-architecture.

Tan & Le (2019) focus solely on searching networks' macro-architecture/outer-skeleton (depth, width and input resolution) while using a pre-determined chain-structured architectural topology with fixed micro

Table 2: An overview of search spaces containing macro-architecture search variables.

| NAS Method | Macro Search Space Architectural Variables | | | | | | | | |
|---|---|---|---|---|---|---|---|---|---|
| | Depth (Layers) | Width (Channels) | Operation Type | Kernel Size | Strides | Pooling Layers | Fully Connected Layers | Skip Connections | Input Resolution |
| Elsken et al. (2017) | ✓ | ✓ | | ✓ | | | | ✓ | |
| Zoph & Le (2016) | ✓ | ✓ | | ✓ | ✓ | | | ✓ | |
| Real et al. (2017) | ✓ | ✓ | | ✓ | ✓ | | | ✓ | |
| Kandasamy et al. (2018) | ✓ | ✓ | | ✓ | ✓ | ✓ | ✓ | ✓ | |
| Cai et al. (2018) | ✓ | ✓ | | ✓ | ✓ | ✓ | ✓ | | |
| Baker et al. (2016) | ✓ | ✓ | | ✓ | ✓ | ✓ | ✓ | | |
| Siddiqui et al. (2025) | ✓ | ✓ | ✓ | ✓ | | | | | |
| Cai et al. (2020) | ✓ | ✓ | | ✓ | | | | | ✓ |
| Tan & Le (2019) | ✓ | ✓ | | | | | | | ✓ |
| Berman et al. (2020) | ✓ | | | | | | | | |
| Sahni et al. (2021) | ✓ | ✓ | | | | | | | |
| Ours | ✓ | ✓ | | | | | | | |

variables. Interestingly, without searching for micro variables, various model variants, from EfficientNet-B0 to EfficientNet-B7, achieve varying Top-1 ImageNet accuracy, i.e. from 77.1% to 84.3%, achieving the (then) state-of-the-art results. Moreover Siddiqui et al. (2025), also reports most of the performance gains using macro search.

AOWS (Berman et al., 2020) has a search space for finding suitable number of channels for each layer, which can get computationally intensive for a larger number of layers, each having different possible channels. The most closely related search space to us is that of CompOFA (Sahni et al., 2021). CompOFA down-samples OFA's (Cai et al., 2020) huge search space based on the effectiveness of compound scaling of depth and width reported by Tan & Le (2019) and quantized-linear-scaling of Radosavovic et al. (2020). However, these studies are conducted on the ImageNet dataset, which has sufficiently high spatial resolution to support exploration of various depth and width scaling rules. While it is possible to search for compound scaling coefficients for datasets with unknown resolutions, some datasets exhibit input dimensions for which scaling laws derived from ImageNet may not hold. For instance, Tan & Le (2019) argue that larger spatial resolutions benefit from deeper networks to capture more complex features. In contrast, as shown in Table 1, the Conway dataset has a resolution of only $10 \times 10$, where a wider rather than deeper architecture may be more appropriate. A similar observation may apply to the Windspeed dataset, whose highly asymmetric input dimensions ($6 \times 768$) differ substantially from those of ImageNet. We therefore allow the search to adaptively explore both depth and width, rather than imposing compound or linearly quantized scaling rules. Instead, our approach progressively evaluates candidate architectures with increasing model capacity. Section 3.1 introduces the proposed macro search space.

## 2.2 Search Strategy

Broadly, NAS search strategies fall into two main categories; black box optimization and one-shot techniques. The most notable black box NAS search strategies are based on reinforcement learning (RL) (Zoph & Le, 2016; Baker et al., 2016; Cai et al., 2018), evolutionary algorithms (Real et al., 2017; Suganuma et al., 2017; Elsken et al., 2017; Lopes & Alexandre, 2023) and bayesian optimization (BO) (Falkner et al., 2018). In general, black-box optimization techniques are computationally more demanding than one-shot techniques due to independent architectural evaluations, however, they are more robust and generalizable to new datasets. One-shot methods offer tremendous speed-ups, however, weight sharing across architectures is known to have several issues, the most notable of which is unstable relative rankings (Zela et al., 2020; Yu et al., 2019; Zhang et al., 2020).

Most of the black box methods evaluate all candidates under one training hyperparameter setting, however, various works have highlighted that training protocol plays a significant role in achieving higher accuracy and different architectures may require different hyperparameters (Yang et al., 2019; Bansal et al., 2022; Zela et al., 2018). Hence, a fixed training setting may favor some models over others leading to premature elimination of potential candidates. Moreover, since both architecture and training protocol is unknown, this

effect may further pronounce for unseen data. Therefore, we opt for a search algorithm which progressively increases the evaluation budget for promising candidates. Our search algorithm therefore is most closely related to multi-fidelity search strategies proposed by Jamieson & Talwalkar (2016), Li et al. (2018) and Falkner et al. (2018).

Jamieson & Talwalkar (2016) propose a simple search strategy with Successive halving (SH). Candidates are first evaluated for a given initial budget. The lowest performing half is then discarded, and the budget is doubled for the remaining candidates, and this process is repeated iteratively until only one candidate remains. SH however, requires pre-determining a trade-off between the number of configurations evaluated and the budget allocated to each. With a fixed total budget, using smaller budgets may prematurely discard strong candidates, while larger budgets may waste resources on poor configurations. To solve this problem, Li et al. (2018) propose Hyperband (HB) which divides the total budget into several combinations of number of configurations vs. budget and run SH for each. Some configurations run on maximal budgets while others on lowest. This suffers from the same issue of some models having an unfair advantage over others and may result in inaccurate relative rankings. Moreover, both SH and HB use randomly sampled configurations that may lead to worse final performance. (Falkner et al., 2018) propose to use bayesian optimization with Hyperband (BOHB) instead of random search and demonstrates strong anytime as well as final performance. BOHB, however, requires building a surrogate model during search which adaptively improves with the number of candidate evaluations. It is possible to apply BOHB for unseen data when there is a 'good' number of evaluations, but in case of larger data, the time constraint may not allow a large number of evaluations and a surrogate model with fewer evaluation points may not be accurate.

Although similar in principal to methods mentioned above, we propose a dataset-adaptive and architecture-aware fidelity budget. Moreover, instead of using random search (Jamieson & Talwalkar, 2016; Li et al., 2018) or surrogate performance predictors (Falkner et al., 2018), we use local search to collect an initial pool of promising candidates, since it has been shown to be a fast and competitive baseline for NAS (Den Ottelander et al., 2021; White et al., 2021a).

## 2.3 Performance Estimation

To identify the optimal architecture, early NAS approaches followed the traditional train/validate paradigm, but the large search spaces they explored required thousands of GPU days (Zoph & Le, 2016; Zoph et al., 2018; Real et al., 2017). To reduce the evaluation cost, NAS methods often rely on low-fidelity proxy estimates, such as using fewer training epochs as in learning curve extrapolation methods (Domhan et al., 2015; Yan et al., 2021; Baker et al., 2017), or training on small data subsets (Klein et al., 2017; Prasad et al., 2024; Shim et al., 2021; Na et al., 2021). However, unseen datasets may exhibit varying convergence rates, making it difficult to predefine a fixed early-termination criterion that generalizes reliably across datasets. Moreover, the achievable accuracy is unknown *a priori*; therefore, the criterion for selecting a representative subset does not exist to begin with. Other low-fidelity estimation methods use reduced input resolutions (Chrabaszcz et al., 2017), or searching with decreased network width (Zoph et al., 2018). While these approximations greatly lower computation, large fidelity gaps can significantly alter rankings (Zela et al., 2018). We discuss these further in Section 3.2.

Another line of research is zero-cost proxies (Krishnakumar et al., 2022), which estimate an architecture's performance with minimal or no training. Such decoupling of architectures from target dataset contradicts the fundamental goal of NAS; discovering dataset specific architectures. Moreover, studies such as (Chen et al., 2021a; Ning et al., 2021) show that proxies developed for modular search spaces do not transfer well to macro spaces.

Some works reuse learned weights by transferring to architecturally similar candidates to speed up evaluation (Real et al., 2017; Chen et al., 2016). However, this gives an unfair advantage to candidates evaluated later in the search process. A similar idea is that of using a supernet, first introduced by Saxena & Verbeek (2016) and later popularized by Liu et al. (2018), where subgraphs inherit weights from a large over-parameterized network. However, Yu et al. (2019) demonstrate that weight sharing can produce inaccurate rankings. Another strategy is to use model-based accuracy predictors (Wen et al., 2020; White et al., 2021b) but these require architecture–accuracy paired data, which itself is costly to acquire.

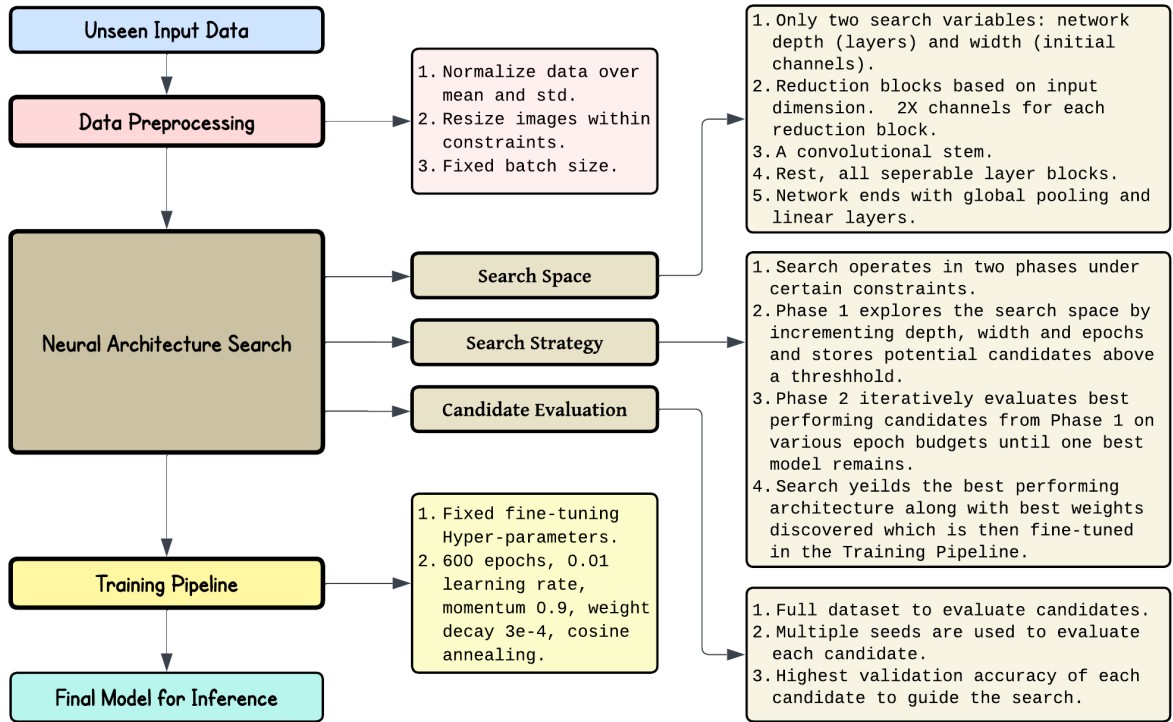

Figure 1: An overview of the NAS Unseen-Data Pipeline. Data Pre-processing and Training Pipeline is fixed for all datasets. Our contribution lies in the Neural Architecture Search block: a minimal yet flexible search space, a multi-fidelity search strategy and a high-fidelity performance estimation.

In this work, we favor rigorous evaluation of a compact yet architecturally diverse candidate set over shallow assessment of an expansive but largely homogeneous architecture space. We describe our evaluation strategy in Section 3.2.

## 3 Methodology

An overview of a standard NAS pipeline, along with the summary of our proposed framework, is shown in Figure 1. Since the datasets are assumed to be unseen, we employ only broadly applicable preprocessing techniques such as normalization and image resizing. Likewise, under strict time constraints, we fix the final training hyperparameters across all datasets and focus solely on NAS, although joint optimization of hyperparameters and architectures has been shown to improve performance (Bansal et al., 2022; Zela et al., 2018). Next, we describe our search space design followed by performance estimation methodology and search strategy.

### 3.1 Search Space Design

As discussed in Section 2.1, we focus exclusively on macro-only search due to its sufficient representational capacity and reduced search complexity. More specifically, we limit the search to just two integer-valued variables, i.e. number of layers (depth) and number of initial channels (width) of the network. Next, we describe why and how certain search variables shown in Table 2 can effectively be removed from the design space.

### 3.1.1 Macro-level Hyperparameters

We limit the depth to a maximum of 100 layers i.e. not targeting very deep architectures and propose to have a simple chain-structured architecture where a network is a sequence of layers without any branching layers, therefore, we drop *Skip Connections* variable from the design space. Moreover, we confine the channel search to initial layer with a fixed channel-doubling rule whenever the spatial resolution halves, as in Liu et al. (2018). Table 2 shows that Tan & Le (2019) is the only one searching for the resolution under the intuition that higher-resolution data require deeper and wider networks to capture complex patterns. While resolution search provides a decent accuracy-latency trade-off for ImageNet dataset, in practice, unseen-data can exhibit arbitrary resolutions and it is not something we can control or anticipate. Moreover, for the core objective of achieving higher accuracy, it is desirable to leverage the available spatial information without excessive resampling. The reason is that up-scaling does not add more to information than there already is but substantially increases network size and hence computational cost, while down-sampling may degrade certain clues in the data. Moreover, natural dataset dimensions can effectively be utilized to calculate a suitable number of reduction blocks. This simple technique offers two advantages: 1) a dataset adaptive sub design space is crafted within suitable macro-level bounds, that is, search only what might be effective, and 2) a critical aspect of macro-architecture, that is, the number of reduction blocks, is 'discovered' prior to search, as opposed to a manually determined fixed number as in Zoph et al. (2018), Liu et al. (2018), and Siddiqui et al. (2025) or explicitly searching for these as in Baker et al. (2016). Therefore, we adopt the native resolution and hence drop *Resolution* search variable.

### 3.1.2 Micro-Architecture Hyperparameters

Following Siddiqui et al. (2025), we use FCN like architectures (Long et al., 2015), replacing fully-connected layers with global pooling and pooling layers with stride-2-convolutions for reduction layers (stride-1 otherwise). It should be noted that global pooling size changes with the number of channels in the final feature map, hence, it is still flexible to accommodate varying datasets. Moreover, for reducing spatial image dimensions, stride-2-convolution is better than pooling due to learnable parameters. These design decisions drop micro-hyperparameters *Strides*, *Pooling Layers* and *Fully Connected* from our search space. Further, as shown in Table 2, nearly all spaces include kernel size search although a 3×3 kernel offers an effective accuracy-computational trade-off. Moreover, Siddiqui et al. (2025) reports only marginal gains during kernel and operation search (i.e. 95.82% vs. 96%) while possessing substantial performance variance across the full search space (76.11% for the worst and 94.65% for the best randomly sampled networks). Therefore, following Tan & Le (2019), we fix *Kernel Size* as well as *Operation Type* variables. However, our layer is composed of ReLU-Drop-Seperable3x3-BN instead of mobilenet/resnet blocks in Tan & Le (2019). We assume this is a transferable cell/block/layer across different datasets.

To this end, we propose a compact search space defined by two independently acting variables, ***layers*** and ***channels***, as summarized in Table 2. When combined with dataset-dependent input resolutions, searchable network depth and width induce substantial non-compound architectural diversity (as opposed to CompOFA Sahni et al. (2021)), yielding a flexible yet expressive design space for varying unseen-data. With these search variables defined, we can now formally state the architecture search problem.

### 3.1.3 Search Problem

Let $\mathcal{D}$ denote a dataset with fixed input resolution $R_\mathcal{D} \in \mathbb{N}^2$. An architecture is denoted by $x \triangleq (L, C)$, where $L \in \mathbb{N}$ and $C \in \mathbb{N}$ represent the number of layers and channels, respectively. For a given dataset $\mathcal{D}$, we define the dataset-specific architecture search space as

$$\mathcal{X}_\mathcal{D} \triangleq \mathcal{X}_\mathcal{D}(R_\mathcal{D}) = \{(L, C) \mid L \in [L_{\min}, L_{\max}]_{\Delta_L}, \ C \in [C_{\min}, C_{\max}]_{\Delta_C}\}, \tag{1}$$

where $\Delta_L$ and $\Delta_C$ specify the discretization step sizes for $L$ and $C$, respectively, inducing finite discrete sets of admissible layer and channel counts within the bounds $[L_{\min}, L_{\max}]$ and $[C_{\min}, C_{\max}]$. The dependence on $R_\mathcal{D}$ reflects that admissible architectural variations are naturally bounded by the spatial resolution of the dataset.

For an architecture $x \in \mathcal{X}_\mathcal{D}$ and network parameters $\theta \in \mathbb{R}^{d(x)}$, let $\mathcal{L}_\text{train}(\theta, x; \mathcal{D})$ and $\mathcal{L}_\text{val}(\theta, x; \mathcal{D})$ denote the training and validation losses, respectively. We assume that for every $x \in \mathcal{X}_\mathcal{D}$, the inner optimization problem admits at least one minimizer.

For a fixed architecture $x$, the optimal network parameters are defined as

$$\theta^*(x; \mathcal{D}) \in \arg \min_\theta \; \mathcal{L}_\text{train}(\theta, x; \mathcal{D}). \tag{2}$$

The dataset-specific optimal architecture is obtained by solving the following bilevel optimization problem, in the spirit of Liu et al. (2018):

$$x_\mathcal{D}^* \in \arg \min_{x \in \mathcal{X}_\mathcal{D}} \; \mathcal{L}_\text{val}\big(\theta^*(x; \mathcal{D}), x; \mathcal{D}\big). \tag{3}$$

For the challenge, the search is restricted by time and GPU memory constraints. Let $\text{Params}(x)$ and $\text{Time}(x)$ denote the number of trainable parameters and the computational cost associated with architecture $x$, respectively. Given a maximum allowable model size $P_\text{max}$ and search budget $T_\text{max}$, the feasible architecture set is

$$\widetilde{\mathcal{X}}_\mathcal{D} = \{x \in \mathcal{X}_\mathcal{D} \mid \text{Params}(x) \le P_\text{max}, \; \text{Time}(x) \le T_\text{max}\}. \tag{4}$$

Accordingly, the outer optimization in equation 3 is performed over $\widetilde{\mathcal{X}}_\mathcal{D}$.

### 3.1.4 Search Space Complexity

Since unseen datasets may exhibit varying input resolutions, our search space implicitly accounts for this variability by parameterizing architectures with the triplet $(L, C, R)$, where the resolution $R$ may vary over a finite discrete set $\mathcal{R} \subset \mathbb{N}^2$ across datasets. Let $L_n$ and $C_n$ denote the number of discrete values induced by $[L_\text{min}, L_\text{max}]\Delta_L$ and $[C\text{min}, C_\text{max}]\Delta_C$, respectively, and let $R_n := |\mathcal{R}|$ denote the number of admissible resolutions. The total number of possible architectures in the search space is then $N\text{arch} = L_n \times C_n \times R_n$. In this work, the depth varies between 10 and 100 layers and the channel count between 16 and 2048, with step sizes of 2 and 48, respectively. Moreover, we allow resolutions between $8 \times 8$ and $128 \times 128$, including non-square configurations. The total number of possible configurations is therefore $28.9 \times 10^6$. For comparison, the search space of Liu et al. (2018) contains approximately $10^{18}$ candidates but with limited variance (Yang et al., 2019). For a single given dataset, however, since the resolution is fixed to the dataset's native resolution $R_\mathcal{D}$, the total number of candidates in the search space reduces to $N_\text{arch} = L_n \times C_n$, resulting in 1,978 candidate architectures.

## 3.2 Performance Evaluation

Existing NAS works use a part of training dataset for search, except (Siddiqui et al., 2023; 2025), where some works simply use a random subset of data (Zoph & Le, 2016; Real et al., 2017; Zoph et al., 2018; Liu et al., 2018), while others propose to find a representative data subset (Prasad et al., 2024; Shim et al., 2021; Na et al., 2021). However, this is not generally applicable for the reasons discussed in the subsequent section.

Another, known yet overlooked aspect is effect of random seed on the rankings of the candidates (Yang et al., 2019; Li & Talwalkar, 2020). NAS involves numerous stochastic components, and established best practices proposed by Lindauer & Hutter (2020) and Li & Talwalkar (2020) recommend reporting random seeds used for search and training to ensure reproducibility and fair comparison. However, beyond reproducibility, randomness introduced by stochastic gradient descent (SGD), weight initialization, and data-loader shuffling can cause substantial variation in validation accuracy across different seeds leading to drastic shifting of relative candidate rankings. In the subsequent section, we present and discuss the details of our experiment to study the effect of data subsets and random seeds.

### 3.2.1 Effect of Data Subset and Random Seeds

We conduct a simple experiment where we train a single model with 10 distinct random seeds for 100 epochs. Moreover, we repeat the experiment using 12%, 25%, 50% and 100% random data samples of CIFAR-10.

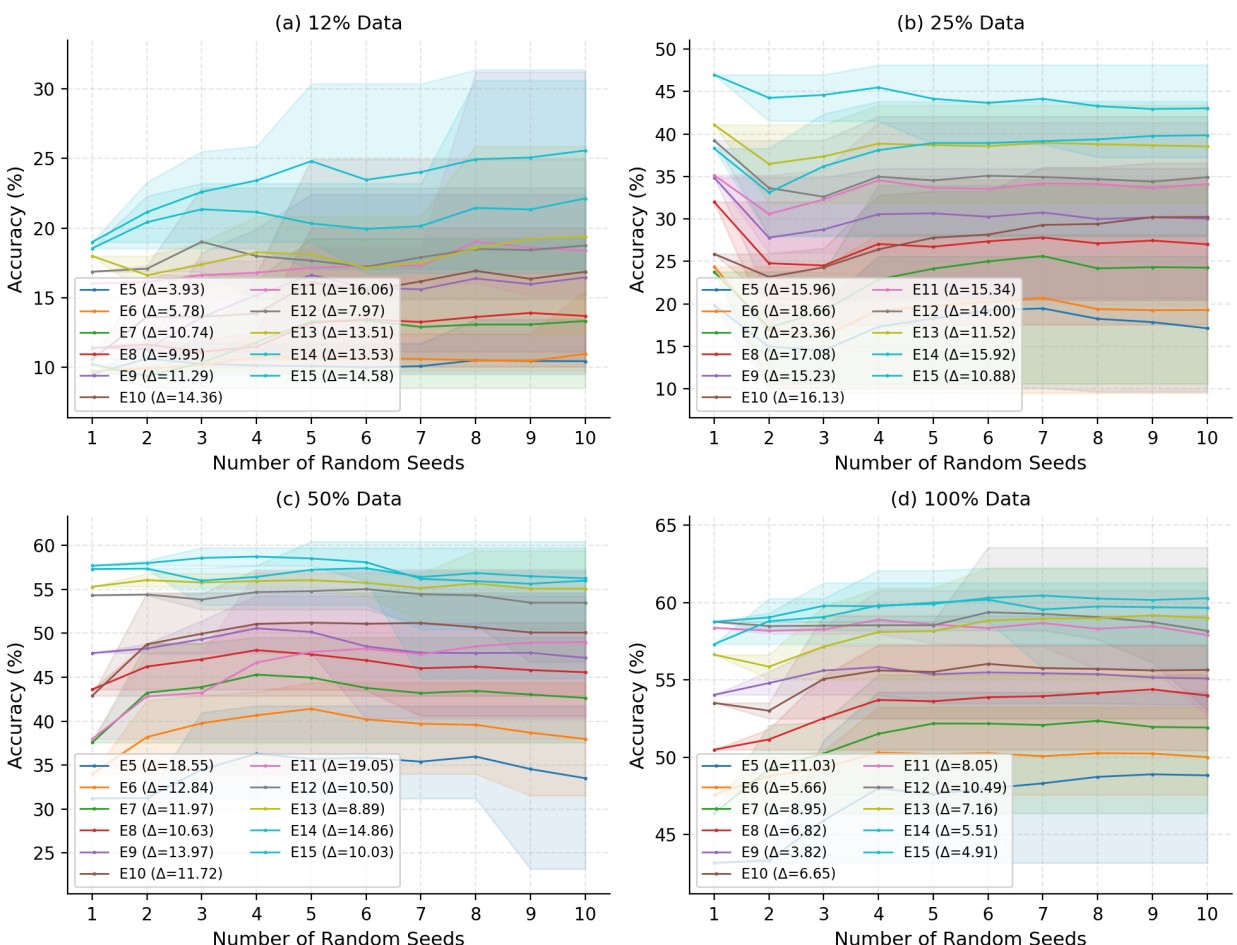

Figure 2: Effect of random seeds at various data and training fidelities. As the number of runs increases with random seeds, new minimum and maximum validation accuracies are observed. Although the performance gap decreases at higher epochs but is still large enough to ignore and fair comparison of architectures under a single seed is questionable.

Figure 2 shows, validation accuracy variance using different subsets of CIFAR-10 data at different epochs. As the number of seeds increase, a new minimum or maximum of validation accuracy is found, enlarging performance gap across all data. This effect is more pronounced for 12%, 25% and 50% subsets with the exception of 12% data at epochs 5 and 6. This is only because the subset is too small to reach higher accuracy. However, when the effect is observed at all epochs collectively, the lowest variance is observed for 100% data with a gradual decrease in variance with epoch increase.

Further, the overlapping mean accuracy lines show that a model achieving a higher accuracy with a certain seed on a particular epoch may produce a lower accuracy at next epochs on different seeds, (e.g. see Figure 2 (d) at epochs E11 and E13). Model performs better at 11 epochs on seeds 1-5, whereas it performs better at 13 epochs on seeds 6-10. Ideally, any model should improve accuracy at increased evaluation budget when this far from saturation (~60% validation accuracy). However, this effect hasn't been captured unless multiple seeds are employed.

These results indicate three important aspects: (1) relying on a single seed and a fixed evaluation budget may hinder accessing the capability of a given model, (2) using 100% of the data generally produces the

least performance variance across seeds and various epochs, (3) with higher evaluation budgets, relatively lower variance can be achieved, however, there is no single best evaluation budget.

We would also like to highlight that even at 100 epochs, the difference across seeds remains fairly high, that is, 6.69%, 6.37%, 4.39% and 5.76%, for 12%, 25%, 50% and 100% data, respectively. Moreover, such high fidelity evaluation with multi-seed is not computationally suitable. Therefore, we have kept our evaluation budget between 1-20 epochs and shown the slightly trimmed version in Figure 2.

To this end, we show that multiple seeds provide candidates with several opportunities to reach their best performance. Therefore, in our experiments, we use five random seeds to evaluate candidates at various epoch budgets and use the highest-seed accuracy. This is because, during search, we are generally navigating models from small to large and increasing evaluation budget, hence ideally expecting a performance increase. Moreover, our core objective is to achieve the highest accuracy and we eventually use the best weights found during search as initial weights for final training, expecting a lucky initialization to yield higher accuracy.

### 3.2.2 Closing the Search-Training Data Domain Gap

Existing works often assume that an architecture discovered using a random or representative subset of the training data will also perform well when trained on the full dataset. This assumption holds only if the subset is truly representative and capable of yielding performance comparable to that obtained using the entire dataset. However, randomly sampled subsets generally violate this assumption, while selecting an appropriate coreset (Prasad et al., 2024; Shim et al., 2021; Na et al., 2021) is itself a challenging problem. Moreover, for unseen datasets, the optimal accuracy achievable on the full training set is unknown a priori, yet this metric is required to identify a representative subset in the first place. In addition, training data are typically augmented during final training to improve performance, implying that a representative subset should also account for such augmentations. Otherwise, an architecture discovered using a subset is optimized for that subset rather than for the final training distribution. Consequently, subset-based search decouples the architecture discovery process from the target dataset, undermining the core objective of NAS: discovering dataset-specific architectures.

In this work, we perform both search and training on the full dataset to ensure that the architecture identified during the search phase is genuinely optimal for the given data and can effectively exploit the entire training set to maximize final performance. An additional benefit of using the full dataset is that warm-up weights obtained during search can be reused during subsequent training. We note that suitable data augmentation strategies are also unknown for unseen datasets and are not optimized in this work; therefore, the provided training datasets are used as is. This evaluation choice is further validated through the extensive ablation study presented in Section 4.3, which confirms that searching with the full training set ultimately yields better-suited architectures than using data subsets (Table 4).

## 3.3 Search Strategy

Using the entire dataset for search as well as evaluating each candidate across multiple seeds aids high-fidelity performance estimates but becomes computationally prohibitive. Therefore, we propose a bi-phase multi-fidelity search strategy as shown in Algorithm 1. Phase 1 progressively expands architectures and increases evaluation budget only as required by the dataset, whereas Phase 2 uses an always-increasing, unique, candidate-aware budget to enforce an equal opportunity across candidates.

### 3.3.1 Phase 1: Progressive Architectural Expansion

The search begins in Phase 1 using a baseline model's validation score. The algorithm then explores the search space by first increasing network depth and subsequently increasing width. Depth or width is expanded as long as each modification yields an improvement in validation accuracy. Any candidate that surpasses the current best score is retained for further evaluation in Phase 2. When neither depth nor width leads to additional gains, the training budget is increased by one epoch, and depth–width exploration resumes. This process repeats until reaching either a time limit or a parameter budget. Phase 1 concludes

---

**Algorithm 1:** NAS Search Algorithm

---

**Input:** Dataset $\mathcal{D}$, search budget $T$, parameter budget $P_{\max}$
**Initialization:**
Allocate search and train budgets $(T_1, T_2, T_{\text{train}})$, initialize architecture and evaluation variables
$(L, C, E)$, and candidate list $\mathcal{L} \leftarrow \emptyset$, train baseline architecture and obtain $Acc_{\text{best}}$

**Phase 1: Architecture Expansion**
**while** $t < T_1$ **and** $P < P_{\max}$ **do**
    |   Expand depth while accuracy improves; add candidate to $\mathcal{L}$ and update $Acc_{best}$
    |   Expand width while accuracy improves; add candidate to $\mathcal{L}$ and update $Acc_{best}$
    |   Increase evaluation budget $E$ if no improvement

**Phase 2: Iterative Elimination**
**while** $|\mathcal{L}| > 1$ **and** $t < T_2$ **and** $E < E_{\max}$ **do**
    |   Sort $\mathcal{L}$ by model size
    |   Evaluate each smaller model with $E \leftarrow E + 1$
    |   Eliminate candidate if no improvement else update $Acc_{best}$

**return** *Architecture and weights for $Acc_{best}$*

---

with a pool of promising candidates, each identified as the best-performing configuration for its respective depth–width–epoch setting.

### 3.3.2 Phase 2: Iterative Multi-Fidelity Candidate Refinement

Phase 2 operates iteratively by evaluating models under progressively increasing training budgets. Candidates are first sorted in ascending order by parameter count, and each smaller model is granted one additional training epoch. This provides lightweight models with improved learning opportunities, mitigating concerns that their performance may appear inferior simply due to insufficient training budgets. As in Phase 1, any candidate that exceeds the current best validation score is retained for further evaluation. After all Phase 1 candidates have been assessed, they are resorted by parameter count, and the iterative evaluation continues until all but one candidate have been eliminated. If the search reaches the time limit before convergence, the model with the highest validation accuracy so far is selected as the final architecture. We observe best anytime as well as the best final performance for the challenge datasets, as discussed in Section 5.5.

## 4 Experiments and Results

Our work primarily focuses on improving the generalization of NAS beyond standard benchmarks (details presented in Section 5). However, a large body of NAS research relies on the CIFAR-10 dataset (Krizhevsky et al., 2009); therefore, to enable comparison with existing methods, we evaluate our framework on CIFAR-10, which consists of 60,000 images with a spatial resolution of $3 \times 32 \times 32$. These images are evenly distributed across 10 classes, with 50,000 used for training and 10,000 for testing.

Moreover, while our primary focus is macro-search, we also evaluate our approach within the widely used DARTS search space (Liu et al., 2018) to facilitate fair comparison with broader NAS literature. Accordingly, this section is organized into two parts: Section 4.1 presents the details and results of our proposed **macro-search** framework, while Section 4.2 examines the transferability of our approach to the **micro-search** (DARTS) search space. Section 4.3 provides details of various ablation studies.

### 4.1 Macro-Search Experiments

We conduct macro-search experiments under two different settings. In the first setting, a layer is defined as a simple ReLU–Dropout–Separable-3×3–BN block, as described in Section 3.1. We refer to this as the *Plain-layer* setting. In the second setting, a layer is defined as a cell, following (Liu et al., 2018). Here, we

Table 3: Comparison with state-of-the-art NAS architectures for CIFAR-10. Top block presents methods supporting macro-architecture discovery while bottom block presents modular (cell-based) architecture search methods.

| NAS Method | Test Err. (%) | Params (M) | Search Cost (GPU-days) | Search Space | Search Algorithm |
|---|---|---|---|---|---|
| **NAS-RL (Zoph & Le, 2016)** | 3.65 | 37.4 | 22400 | Macro | RL |
| **Meta-QNN (Baker et al., 2016)** | 6.92 | 11.2 | 100 | Macro | RL |
| **EAS (Cai et al., 2018)** | 4.23 | 23.4 | 10 | Macro | RL |
| **Evolution (Real et al., 2017)** | 5.40 | 5.4 | 2600 | Macro | EA |
| **Genetic-CNN (Suganuma et al., 2017)** | 5.98 | 1.7 | 14.9 | Macro | EA |
| **NASH-Net (Elsken et al., 2017)** | 5.20 | 19.7 | 1 | Macro | EA |
| **RandGrow (Hu et al., 2018)** | 3.38 | 3.1 | 6 | Macro | RS |
| **Petridish (Hu et al., 2019)** | 2.83 | 2.2 | 5 | Macro | Gradient |
| **NASBOT (Kandasamy et al., 2018)** | 8.69 | N/A | 1.7 | Macro | SMBO |
| **NSGA-NET (Lu et al., 2019)** | 3.85 | 3.3 | 8 | Macro | SMBO |
| **Macro-NAS (Lopes & Alexandre, 2023)** | 4.23 | 6.7 | 1.03 | Macro | EA |
| **EGNAS (Siddiqui et al., 2025)** | 3.17 | 2.49 | **0.43** | Macro | Greedy |
| **Ours (Plain-layer)** | 3.38 | 3.7 | 0.54 | Macro | Multi-fidelity |
| **Ours (DARTS-layer)** | **2.39** | 10.4 | 1.18 | Macro | Multi-fidelity |
| **NASNet-A (Zoph et al., 2018)** | 2.40 | 27.6 | 2000 | Modular | RL |
| **AmoebaNet-B (Real et al., 2019)** | 2.13 | 34.9 | 3150 | Modular | Evolution |
| **DARTS (Liu et al., 2018)** | 2.76 | 3.3 | 4 | DARTS | Gradient |
| **GDAS (Dong & Yang, 2019)** | 2.82 | 2.5 | 0.17 | DARTS | Gradient |
| **PC-DARTS (Xu et al., 2019)** | 2.57 | 3.6 | 0.1 | DARTS | Gradient |
| **P-DARTS (Chen et al., 2021b)** | 2.5 | 3.4 | 0.3 | DARTS | Gradient |
| **AGNAS (Sun et al., 2022)** | 2.46 | 3.6 | 0.4 | DARTS | Gradient |
| **FP-DARTS (Wang et al., 2023)** | 2.5 | 3.9 | 0.08 | DARTS | Gradient |
| **RF-DARTS (Zhang et al., 2023)** | 2.6 | 4.6 | 0.4 | DARTS | Gradient |
| **IS-DARTS (He et al., 2024)** | 2.56 | 4.25 | 0.42 | DARTS | Gradient |
| **SWAP (Peng et al., 2024)** | 2.48 | 4.3 | 0.004 | DARTS | Training-free |
| **NCD-NWOT (Kang et al., 2025)** | **2.44** | 4.4 | **0.002** | DARTS | Training-free |
| **Ours (Micro-Search)** | 2.69 | 4.2 | 0.94 | DARTS | Multi-fidelity |

perform macro-search using a DARTS-discovered cell used as a layer, and refer to this as the *DARTS-layer* setting.

### 4.1.1 Macro-Search Settings

For *Plain-layer* setting, we run search with $L_{min}$=8, $L_{max}$=100, $C_{min}$=16, $C_{max}$=2048, $L_{res}$=2, $C_{res}$=32, $E_{min}$=1 and $E_{max}$=50. We use the entire CIFAR-10 training set for search and employ 5 random seeds for each candidate evaluation. Further, we restrict the maximum model size to 4.2M parameters. The effect of using smaller subsets of data for search are discussed in the Section 4.3.

For *DARTS-layer* setting, since a single layer contains significantly more parameters than a *Plain-layer*, we reduce the channel resolution and increase the parameter budget to allow evaluation of a reasonable number of candidates during search. More specifically, we set $C_{min}$ and $C_{res}$ to 8 and 16, respectively and increase the maximum parameter budget to 12.5M.

### 4.1.2 Evaluation and Training Settings

For both search and training, we follow the settings of Liu et al. (2018); Xu et al. (2019); Chen et al. (2021b), including standard augmentations such as random crop, horizontal flip, and cutout. All candidates are trained using SGD with momentum 0.9, weight decay of 3e-4, and an initial learning rate of 0.025. However, during search, the learning rate is fixed at 0.025 and is not decayed as in final training, and the number of epochs varies adaptively across candidates according to our multi-fidelity search strategy, unlike the fixed 600 epochs used for final training.

For final training of the *Plain-layer* network, we use the best weights discovered during search. In contrast, for the *DARTS-layer* network, we follow the exact training code and settings of Xu et al. (2019) (training from scratch), modifying only the number of layers and channels as discovered by our method.

### 4.1.3 Macro-Search Results

The top block of Table 3 presents a comparison with several macro-NAS methods. We achieve a competitive test error of **3.38%** (best across 10 runs) with the *Plain-layer* setting, while obtaining the lowest error of **2.39%** (with three search runs yielding the same model twice, and thus trained only once) using the *DARTS-layer* setting. This result is the lowest among all search spaces except AmoebaNet (2.13%, second block), which is orders of magnitude more expensive (approximately 3150 GPU days) and employs a modular search space that differs from DARTS. While direct comparison across studies should be interpreted with caution due to differences in search spaces and optimization strategies, the results nevertheless provide useful context for assessing the effectiveness of our approach relative to prior macro-NAS methods.

It should also be noted that our results are better than all methods using the DARTS search space (bottom block), even though we simply employ a DARTS (second-order) cell. Interestingly, for the *DARTS-layer* setting, our search algorithm discovers a 16–64 (layers–channels) configuration, compared to the default DARTS setting of 20–36, highlighting the importance of macro-search.

Using a single GPU[2], the search takes a maximum of 0.54 and 1.18 GPU days for the *Plain-layer* and *DARTS-layer* settings, respectively. For the *Plain-layer* setting, a total of 33 candidate evaluations are performed across different budgets during search, with 19 and 16 models evaluated in Phase 1 and Phase 2, respectively. However, the number of unique models within the 4.2M parameter constraint is limited to 16, with promising candidates re-evaluated under different budgets ranging from 2 to 21 epochs. The evaluation cost varies across candidates due to adaptive fidelity assignment.

For the *DARTS-layer* setting, a total of 24 candidate evaluations are performed, with 14 and 12 models evaluated in Phase 1 and Phase 2, respectively. The 12.5M parameter constraint limits the number of unique models to 12. Promising candidates are evaluated for 2 to 16 epochs.

## 4.2 DARTS-Search Experiments

To isolate the contribution of the proposed search strategy and ensure a fair comparison with existing micro-search methods, we adopt the well-studied and widely used DARTS search space (Liu et al., 2018). The DARTS search space enables cell-level architecture discovery while keeping the macro-architecture fixed at 20 layers (cells) and 36 initial channels for CIFAR-10. Since our search strategy is designed to explore depth and width configurations through the layers and channels variables, we make some adjustments to apply it to the DARTS search space.

### 4.2.1 Macro to Micro Search

As described in Section 3.3, our macro-search strategy begins Phase 1 with a small baseline model and explores network depth and width alternately in increasing order. Whenever validation accuracy fails to improve, the evaluation budget is increased by allocating additional training budget. This process yields a pool of promising candidates, where each successive candidate is larger in terms of learnable parameters. Candidates that fail to outperform the current best model are discarded.

For the DARTS search space, however, the macro-architecture is fixed, making direct depth-width exploration infeasible. Instead, we use the number of learnable parameters as the primary search variable. To construct a suitable search space, we first determine the smallest and largest networks achievable within the DARTS framework by appropriately configuring the cell structure. The smallest network is obtained by selecting a single learnable operation, namely *separable-3×3*, in the normal cell while assigning all remaining connections in both normal and reduction cells to either *skip* or *none* operations. Conversely, the largest network is constructed by assigning the *separable-5×5* operation to all connections in both normal and reduction cells.

---

[2]All experiments are conducted on an Nvidia Quadro RTX 8000

Table 4: Accuracy and search cost trade-off when using different subsets of data.

| | Macro-Search | | | | | | Micro Search | | |
|---|---|---|---|---|---|---|---|---|---|
| | Plain-Layer | | | DARTS-Layer | | | | | |
| Dataset Size | Model Size Params (M) | Accuracy | Search Cost GPU Hours | Model Size Params (M) | Accuracy | Search Cost GPU Hours | Model Size Params (M) | Accuracy | Search Cost GPU Hours |
| **D-12** | 1.33 | 95.23 | 1.8 | 0.15 | 94.16 | 2.67 | 1.9 | 96.80 | 4.6 |
| **D-25** | 1.95 | 95.94 | 4.9 | 1.38 | 97.22 | 11.3 | 2.9 | 97.28 | 9.2 |
| **D-50** | 2.13 | 95.93 | 6.5 | 12.2 | 97.55 | 20.4 | 2.9 | 97.28 | 14.1 |
| **D-100** | 2.71 | **96.30** | 13 | 10.4 | **97.61** | 28.3 | 4.2 | **97.31** | 22.7 |

We then generate a fixed number of candidate architectures by randomly sampling cells whose parameter counts are approximately equally spaced between these two extremes. This procedure produces a compact yet diverse search space in which cell-level variations substantially influence the resulting architecture despite the fixed macro-architecture. Similar to macro-search, when a candidate fails to improve upon the current best model, its evaluation budget is increased by one additional epoch. Phase 2 of the search algorithm remains unchanged.

### 4.2.2 Micro-Search Settings

We fix the number of layers and channels to 8 and 16, respectively, following the setup used by (Liu et al., 2018; Xu et al., 2019) for constructing a super-network for search. We restrict the maximum number of equally spaced candidate networks to 16, excluding the smallest and largest networks. We use the entire CIFAR-10 training set for search and evaluate each candidate using five random seeds. Furthermore, the default configuration with 20 layers and 36 channels corresponds to a maximum model size of 5.6M parameters.

### 4.2.3 Evaluation and Training Settings

The evaluation and training settings remain the same as in the macro-search setting. For final training, we use the cell discovered by our algorithm and follow the exact code and settings of Xu et al. (2019). It should be noted that the layers and channels are fixed at 20–36 for this experiment.

### 4.2.4 Micro-Search Results

The search takes 0.94 GPU-days for this setting. The bottom block of Table 3 presents a comparison with several DARTS-related methods. We discover a cell that produces a model of size 4.2M and achieves a competitive error rate of **2.69%**. It should be noted that the DARTS search space contains approximately $10^{18}$ possible cell configurations, whereas we sample only 16 candidate cells based on their learnable parameters and resulting model sizes. In this search, some models with fewer than 3M parameters exhibit error rates above 3% (see Table 4). Nevertheless, our proposed search strategy consistently identifies better-performing architectures, demonstrating its effectiveness within a smaller yet diverse search space.

### 4.3 Ablation Studies

In this section, we investigate the effect of using data subsets on the architectures discovered and the associated performance trade-offs. We present results for both macro-search and micro-search.

### 4.3.1 Models Discovered Using Data Subsets

For the *Plain-layer* macro-search setting, we perform 10 search runs using 12%, 25%, 50%, and 100% of the CIFAR-10 training data, denoted as D-12, D-25, D-50, and D-100, respectively. All other settings remain identical to those in Section 4.1.1. Table 4 (left) reports the mean model size and test accuracy across the 10 runs. While smaller subsets reduce search cost, they also degrade final accuracy. Moreover, larger subsets consistently lead to the discovery of larger architectures, demonstrating the dataset-adaptive nature of the proposed search strategy. A similar trend is observed for the *DARTS-layer* setting (Table 4, center). The

only exception is D-50, which discovers a 12.2M-parameter model in one of the three runs; despite being the largest architecture, it does not achieve the highest accuracy.

For the micro-search experiments, among the 16 candidate cells (G1–G16, ordered by increasing parameter count), the D-12, D-25, D-50, and D-100 settings consistently select genotypes G1, G5, G5, and G11, respectively, across all three runs. Thus, the dataset-adaptive trend observed in macro-search also extends to the cell-based search space. These results indicate that the proposed search strategy, effectively adapts to a given dataset, and that using the entire dataset during search yields the best architecture.

### 4.3.2 Search Cost

The search cost reported in Table 4 is for each candidate evaluated with 5 random seeds. Assuming the search will roughly run the same track, the search cost can be reduced 5X using a single seed. However, the discovery of the best final architecture is can not be guaranteed.

## 5 Unseen-Data Challenge Results and Analysis

This Section presents details of our winning solution for the 5[th] Unseen-Data Challenge which was conducted as part of the AutoML conference 2025. Section 5.1 gives an overview of the competition and its rules, Section 5.2 describes the unseen-datasets introduced in the latest challenge (2025), Section 5.3 describes search and training settings, Section 5.4 discusses the challenge results and Section 5.5 presents post-challenge analysis.

### 5.1 Challenge Details

#### 5.1.1 Overview

The objective of the challenge is to design an end-to-end pipeline comprising data preprocessing, neural architecture search, and final training that generalizes out-of-the-box to diverse unseen datasets. The annual challenge is conducted in three stages. Stage 1, beginning in May, familiarizes participants with the challenge starter toolkit and datasets from previous challenge editions (Geada et al., 2024). In Stage 2, participants' solutions are evaluated over a shorter duration to verify compliance with the challenge requirements before advancing to the final 24-hour evaluation in Stage 3. Importantly, the Stage 2 datasets are used solely to validate submission correctness and are unrelated to the unseen datasets used in the final Stage 3 evaluation. Stage 3 begins in August, and the final results for the latest unseen datasets are announced in September.

#### 5.1.2 Rules

The challenge imposes two key requirements: (1) participants must strictly adhere to a 24-hour time budget across all three datasets, and (2) submissions must run on a single GPU while adapting to the available memory. Solutions that violate either the time or memory constraints are penalized.

### 5.2 Datasets

Table 1 summarizes the three datasets introduced in the 2025 NAS Unseen-Data Challenge, each differing substantially in the number of output classes, input resolution, and the sizes of the training, validation, and test splits. The Conway dataset (Towers et al., 2025b) is a synthetically generated benchmark derived from John Horton Conway's Game of Life, where the objective is to predict the number of living cells in the subsequent generation. Owing to its relatively straightforward underlying rules, this task is expected to be easier for both humans and machine learning models. In contrast, the Cryptic dataset (Ericsson et al., 2025a) focuses on language understanding through the resolution of cryptic clues, a task that is challenging even for humans and represents a moderate level of difficulty. The Windspeed dataset (Ericsson et al., 2025b) requires predicting average wind speed from temperature and pressure measurements and constitutes the most challenging benchmark among the three, with the baseline achieving only approximately

Table 5: NAS Unseen-Data 2025 Challenge Results: Baseline scores are provided by the organizers on a manually designed network. Raw Acc. represents accuracy achieved on the held-out test set, whereas Adjacent (Adj.) Score represents score relative to the baseline. Official leaderboard can be found at: NAS Unseen-Data Challenge 2025 Leaderboard

| Solutions | GameOfLife | | WindSpeed | | Cryptic | | Final Score |
|---|---|---|---|---|---|---|---|
| | Raw Acc. | Adj. Score | Raw Acc. | Adj. Score | Raw Acc. | Adj. Score | |
| Basline | 41.0 | - | 13.492 | - | 71.104 | - | - |
| 3rd-Place | 99.97 | 9.995 | 13.776 | 0.033 | 72.265 | 0.402 | 10.430 |
| Runner-up | 99.66 | 9.942 | 15.278 | 0.206 | 73.235 | 0.737 | 10.885 |
| Winner (Ours) | **99.99** | **9.998** | **17.375** | **0.449** | **76.145** | **1.745** | **12.192** |

13.5% accuracy. Collectively, these datasets span a broad range of difficulty levels, enabling assessment of a method's ability to adapt to diverse real-world scenarios.

Beyond differences in task complexity, the datasets vary considerably in their input dimensions and the number of available training and validation samples. Such variability complicates the design of a robust evaluation protocol and the reliable ranking of candidate architectures under limited search and training budgets. Furthermore, dataset-specific preprocessing strategies and post-search training protocols are unknown a priori for these unseen tasks and remain important directions for future investigation.

### 5.3 Search and Training Settings

A total budget of 24 hours was available for all three challenge datasets. We allocated 8 hours to each dataset, dividing the budget equally between search and final training (4 hours each). Within the search stage, approximately 40% and 60% of the time budget were assigned to Phase 1 and Phase 2, respectively. Additionally, to accommodate datasets with extreme spatial resolutions, input dimensions were clipped to the range of 8×8 to 128×128; datasets exceeding 128 pixels in either dimension were resized to 128×128, while those below 8 pixels were increased to 8×8.

Unless otherwise stated, the search and final training configurations used for the challenge are identical to the *Plain-layer* settings described in Section 4.1.1. The only modifications are a reduced maximum model-size constraint of 3.2M parameters (instead of 4.2M) and a lower initial learning rate of 0.01 during final training (instead of 0.025).

### 5.4 Results

As shown in Table 5, our method outperforms competing solutions across all datasets. The training and validation sets are available during the search and training phases, while the final score is determined using held-out test samples. On the relatively easy GameOfLife/Conway dataset, our framework discovers a near-perfect model, achieving a 99.99% raw accuracy score. On the more challenging Cryptic and Windspeed datasets, it attains 5% and 4% higher raw accuracy than the respective baselines. For the challenge, Geada et al. (2024) propose computing the score for each dataset using $\max(-10, relative\_score)$, where the $relative\_score = (test_{acc} - base_{acc}) * s$ and $s = 10/(100 - base_{acc})$. The final score is obtained by summing the individual dataset scores. This scoring scheme prevents a single dataset from dominating the overall score and instead emphasizes consistent generalization across datasets. Using this metric, our solution achieves a total score of **12.192**, outperforming the second- and third-place solutions, which score **10.885** and **10.430**, respectively, thereby highlighting the strong generalization capability of our framework.

### 5.5 Post Challenge Analysis

In this section, we analyze the framework's robustness to unseen datasets during the actual challenge run.

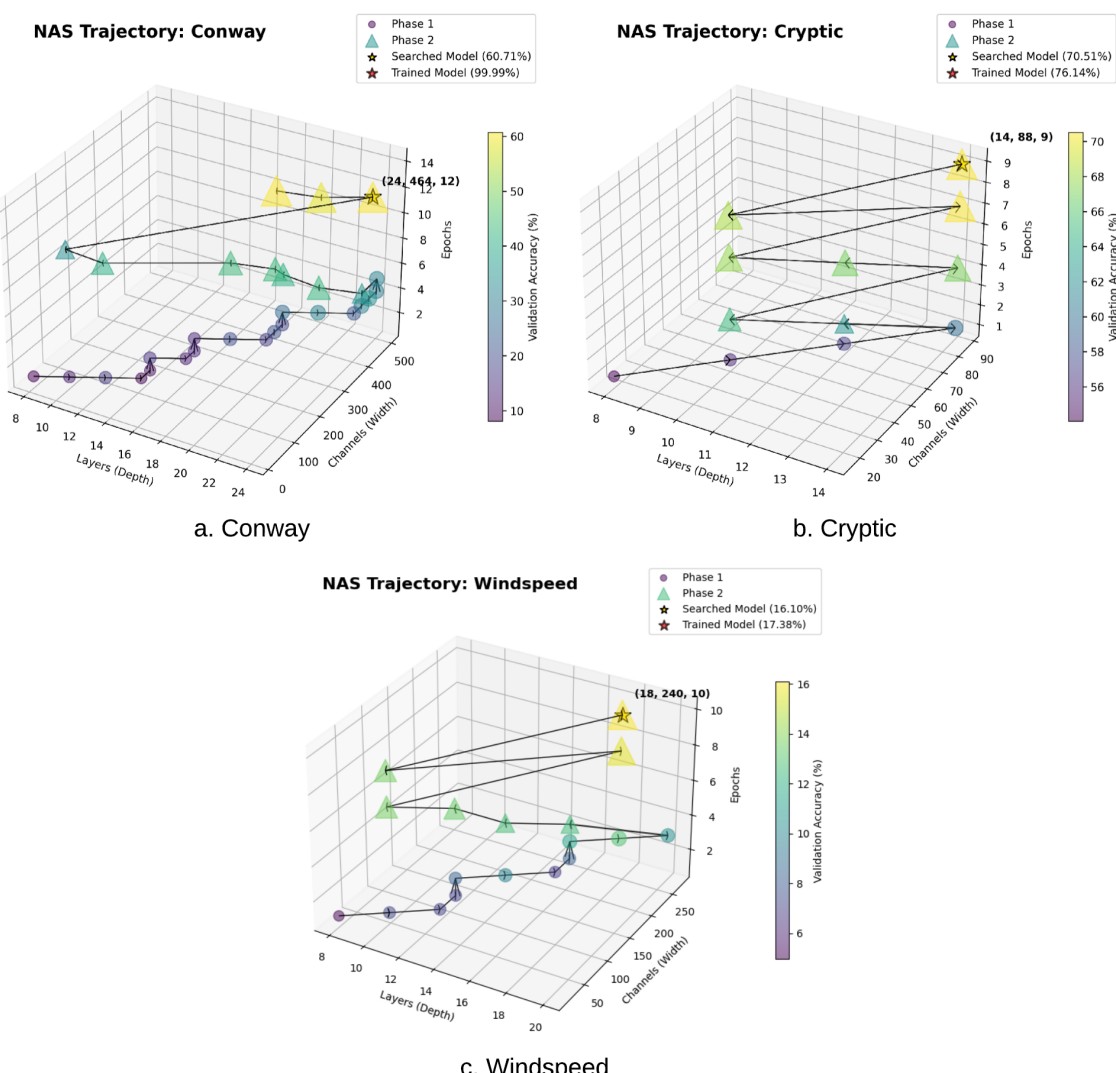

Figure 3: Search Trajectories for Conway, Cryptic and Windspeed datasets during challenge run. Phase 1 evaluation is represented by spheres while Phase 2 using triangles. Moreover, arrows represent the search trajectory and the size of a point represents accuracy. In Phase 1, epochs are increased only when the validation accuracy plateaus, while they are increased for every next candidate evaluation in Phase 2 until a winning model is discovered.

### 5.5.1 Dataset-Adaptive Search

Figure 3 provides a visual overview of the search process for each challenge dataset. Under a strict time budget of 8 hours per dataset, the proposed framework adapts its behavior by dynamically adjusting the allocated search space, number of iterations, and fidelity budget according to dataset characteristics.

Because the Conway dataset has much smaller spatial dimensions than Cryptic ($1\times10\times10$ versus $1\times128\times128$), a larger number of candidates can be evaluated during Phase-1 (20 for Conway compared to 4 for Cryptic). During this phase, both Conway and Windspeed exhibit architecture-level performance

Table 6: Effect of random seed on candidate rankings. C, W and G represent the first three (1,2,3) candidate architectures evaluated for Cryptic, Windspeed and Conway datasets, respectively. The Avg and High rankings are average and highest accuracy across seeds, respectively. The rankings shift arbitrarily across seeds, averaged accuracy and the highest seed accuracy.

| | **Cryptic** | | | | | **Windspeed** | | | | | **Conway** | | | | |
|---|---|---|---|---|---|---|---|---|---|---|---|---|---|---|---|
| **Rank** | | Seed | | Avg | High | | Seed | | Avg | High | | Seed | | Avg | High |
| | 0 | 1 | 2 | | | 0 | 1 | 2 | | | 0 | 1 | 2 | | |
| 1 | C-3 | C-1 | C-2 | C-3 | C-3 | W-2 | W-1 | W-1 | W-3 | W-1 | G-1 | G-1 | G-2 | G-2 | G-2 |
| 2 | C-2 | C-3 | C-1 | C-1 | C-2 | W-1 | W-3 | W-2 | W-2 | W-3 | G-2 | G-2 | G-3 | G-1 | G-1 |
| 3 | C-1 | C-2 | C-3 | C-2 | C-1 | W-3 | W-2 | W-3 | W-1 | W-2 | G-3 | G-3 | G-1 | G-3 | G-3 |

saturation, prompting the allocation of additional training budget, whereas Cryptic continues to benefit primarily from architectural growth. In Phase-2, Cryptic, Windspeed, and Conway undergo 4, 3, and 2 elimination rounds, respectively. Although not shown in the figure, challenge logs indicate that Windspeed and Cryptic reach the model size constraint before exhausting the Phase-1 time budget, demonstrating strong anytime performance. Conway, in contrast, reaches the time limit in both phases but nonetheless achieves the best anytime and final performance.

Throughout the search, competing models vary substantially in size, and smaller architectures are intentionally allocated additional training budget to prevent premature elimination due to insufficient evaluation. This ensures fair comparison across candidates. Moreover, the proposed search space exhibits substantial performance variance despite its compact size: candidate validation accuracies range from 8–60%, 54–70%, and 5–16% for Conway, Cryptic, and Windspeed, respectively. These results confirm that the search space can generate diverse, dataset-adaptive architectures. Moreover, during the search phase, we discover models that already outperform the manually designed baseline architectures on Conway and Windspeed (60.71% and 16.10%, respectively), while performance on Cryptic remains competitive (70.51%). This is achieved without final training. Subsequent full training further improves accuracy across all datasets, with gains varying according to dataset difficulty.

Overall, the results demonstrate that the proposed search framework efficiently adapts to dataset complexity, maintains fair evaluation across architectures, and delivers strong anytime as well as final performance.

### 5.5.2   Multi-Seed Evaluation

For the challenge run, we evaluate each candidate using five random seeds, and Table 6 illustrates the effect of random seeds on candidate rankings. A key observation is that changing the seed leads to substantial variations in the relative rankings of candidates across all datasets, consistent with the findings of Yang et al. (2019). For instance, for Cryptic dataset, model C-3 and C-1 are ranked 1 and 3rd, respectively for seed 0 but vice versa for seed 2. Similar variations can be observed for Windspeed and Conway datasets. Furthermore, rankings obtained by averaging performance across multiple seeds are generally inconsistent with those derived from the best-performing seed, with the exception of the Conway dataset. While the choice between averaged and best-seed performance remains debatable, these results clearly indicate that relying on a single seed yields unstable rankings, even when evaluation is performed on the full dataset. In contrast, multi-seed evaluation captures performance variability and enables more reliable relative ranking of candidates, making it an important consideration for explainable NAS.

### 5.5.3   Multi-Fidelity Search

Multi-seed evaluation also reveals performance variance trends similar to those observed on CIFAR datasets (Section 3.2). For instance, the winning model on the Windspeed dataset exhibits performance variances of 4.84%, 2.73%, and 2.86% across different seeds when evaluated at 3, 8, and 10 epochs, respectively. The winning model for the Cryptic dataset shows variances of 9.22%, 1.84%, 1.34%, and 1.02% at 1, 4, 7, and 9 epochs, respectively. Similarly, the model discovered for the Conway dataset exhibits variances of 15.2% and 13.39% at 4 and 12 epochs, respectively. While performance variance is pronounced at lower training

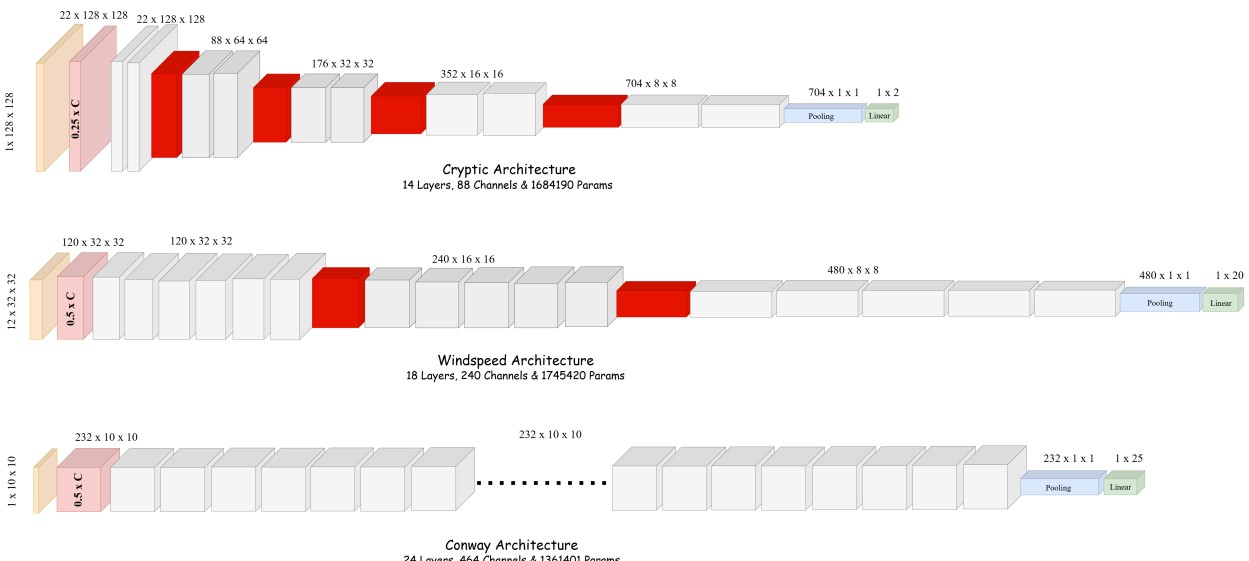

Figure 4: Architectures discovered for the challenge datasets. Each of the three architectures end up with different spatial dimensions, depth, width, global pooling size and number of reduction blocks (shown in red). Block shown in pink is a stem. Note that the stem and first blocks use 0.5×C discovered channels for Conway and Windspeed, and 0.25×C for Cryptic.

fidelities, it decreases substantially as fidelity increases, emphasizing the effectiveness of the proposed multi-fidelity strategy in mitigating variance and stabilizing candidate ranking.

### 5.5.4 Discovered Architectures

Figure 4 illustrates the architectures discovered for the challenge datasets, each characterized by distinct spatial input resolutions, network depth and width, global pooling sizes, numbers of reduction blocks, and total learnable parameters. Notably, no reduction block is selected for the Conway dataset, which has a relatively small input resolution of $10 \times 10$, whereas four and two reduction blocks are assigned to the Cryptic and Windspeed datasets, respectively.

### 5.5.5 Limitations and Future Work

Our work primarily focuses on architecture search, whereas for unseen datasets, other factors such as data augmentation techniques and training hyperparameters (e.g., learning rate and number of training epochs) remain unexplored, presenting a promising direction for future research. Furthermore, the time and GPU memory constraints of the challenge impose limits on the search space, including a parameter budget (e.g., around 3 million) and maximum input image dimensions. As a result, for high-resolution datasets, only a small number of candidate architectures can be evaluated, even though larger models might ultimately prove more effective. Finally, since candidates are trained on the entire dataset during search, both dataset size and image resolution directly impact training time and, consequently, search efficiency. Future work could therefore investigate techniques to accelerate the search process, particularly for large-scale datasets.

## 6 Conclusion

This work addresses limitations of existing NAS methods arising from their heavy reliance on CIFAR, ImageNet, and their derivative benchmarks. Our proposed framework departs from low-fidelity evaluations used to navigate large and complex search spaces, demonstrating that precise evaluation of a small number of candidates leads to stronger NAS generalization on unseen data, and that macro-architecture variations

alone provide sufficient architectural diversity to adapt to datasets with differing characteristics. Moreover, since high-fidelity evaluations are computationally expensive and unseen datasets lack well-defined training protocols, we introduce a novel dual-purpose multi-fidelity search algorithm that both accelerates the search process and evaluates candidates under multiple training budgets, thereby improving robustness across datasets. The proposed framework generalizes well to datasets introduced in the NAS Unseen-Data Challenge and achieves an overall score of 12.19, outperforming the second- and third-place NAS solutions (10.89 and 10.43), respectively. Additionally, it outperforms several macro- and micro-search methods on the widely used CIFAR-10 dataset.

## Acknowledgments

We acknowledge the use of ChatGPT (OpenAI) for language editing and improving the readability of this manuscript.

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
