# OpenReview forum: "NAS Without Priors: A Robust Architecture Search Framework for Unseen-Data"
_TMLR — Accepted by TMLR_

### Review · Reviewer_QUK9 · 2026-03-09

**Summary Of Contributions:**

The authors analyze the limitations of existing NAS practices and propose a framework specifically designed to generalize to unseen data, advocating sparser exploration combined with training on more data and for longer, with multiple random seeding.

**Additional Comments:**

* Fig. 1 is not particularly helpful for a research audience and could safely be eliminated; it doesn't illuminate anything we don't already know about NAS
* The authors use Fig. 3 to attempt to argue that the effect of random seeding is overlooked and significant during NAS; however, the provided evidence is not entirely persuasive on either front. The CIFAR-100 effect is only +- 1.06 after 20 epochs, which is not a very large effect. Furthermore, there is no justification in the paper for why training for just 20 epochs should result in convergence; there's a clear positive convergence trend from 1 -> 5 -> 20, as one would expect, perhaps after 40, 80 or 120 epochs the gap would be still smaller, but how much smaller? And what effect does the size of the random subset play on these trends, versus using the whole dataset?
* Why does Table 1 stop at 2019 for macro-only search methods (Tan & Le), skipping 6+ years of relevant work (RegNet, OFA, CompOFA, AOWS, Differential Model Scaling)?

**Audience:**

No

**Audience Explanation:**

* With a non-standard, undocumented competition carrying the main result and no other experimental comparisons of the method to existing literature, it's hard to see how anyone immersed in this subfield would gauge the significance of this paper's result, and therefore of its proposed method.

**Claims And Evidence:**

No

**Claims Explanation:**

* The authors write: "to the best of our knowledge, no existing NAS work has restricted the search space to just two architectural variables". This is a considerable overclaim and disregards key prior work -- CompOFA (Sahni et al., ICLR 2021) argues that depth and width dimensions should increase or decrease together, not independently (2D), fixing kernel size by default. RegNet (Radosavovic et al., CVPR 2020) argues for a quantized linear relation between model depth and width (effectively 1D search), yielding design spaces with a better concentration of well-performing models. There are also other examples, e.g. EfficientNet.
* The authors write: "This motivation led to the annual NAS Unseen-Data Challenge (Towers et al., 2025a), where participants are required to design a complete NAS pipeline and apply it to three entirely new and previously unseen datasets under a strict 8-hour time constraint per dataset for both search and training. Our proposed approach outperforms manually designed baseline architectures across all unseen datasets and achieves the highest combined score of 12.19, exceeding the second- and third-place solutions, which obtained scores of 10.89 and 10.43, respectively." Despite the fact that this is the only core result in the paper, very little space is devoted to describing / explaining the competition or this result (in fact this is only mentioned in the introduction). The citation points to a URL which has since been updated, so there is no information on the 2025 competition that I can see. Would be helpful to have an archival citation here and also to have a section and perhaps a figure clearly documenting things like the scale of the competition, whether there were other datasets in the competition, how long the authors had to develop solutions, how many submissions allowed, what metrics used, etc.

**Requested Changes:**

CRITICAL
* Re-scope the claim on restricting NAS search space
* Add the relevant related work to Table 1 and the related works section
* Fully document the relevant parameters of the 2025 NAS Unseen-Data Challenge and present the main result in a figure or table, including other competitors and baselines
* Include substantive comparisons to strong, recent methods, using established benchmarks

PREFERRED
* Address the additional comments

---

> ### Author Response · Authors · 2026-04-19
> **Response to Reviewer QUK9**
>
> Thank you for your valuable feedback and pointing us to very relevant literature we had missed.
>
> 1. "Re-scope the claim on restricting NAS search space": We have retracted our claim and updated Sections 3.1 on Search Space Design.
> 2. "Add the relevant related work to Table 1 and the related works section": We have updated Table 2 (Previously Table 1) and Section 2.1 on related work. We also highlighted our reasons for not using compound scaling for unseen data in Section 2.1.
> 3. " Fully document the relevant parameters of the 2025 NAS Unseen-Data Challenge": We have added Section 5.1 about Challenge Details and we are currently working with challenge organizers to add a verifiable link containing challenge details and results from other participants.
> 4. "Include substantive comparisons to strong, recent methods, using established benchmarks": We have added a new section (Section 4) containing comprehensive experiments to compare our work against broader NAS literature for both macro-search and DARTS-based search methods. Section 4.1 is about the contribution of macro-search whereas 4.2 discusses the applicability and effectiveness of the proposed search strategy on DARTS search space, singling out its contribution. We have also added Table 3 that contains comparisons against both macro- and recent micro-search methods and have achieved competitive results.
>
> Additional Comments:
> 1. "Fig. 1 is not particularly helpful" : It has been eliminated.
> 2. "The authors use Fig. 3 to attempt to argue that the effect of random seeding is overlooked and significant during NAS; however, the provided evidence is not entirely persuasive on either front": We have updated Figure 2 (previously Figure 3) to clarify further the effect of multi-seed evaluation and about using data subsets. Further Section 3.2 has been updated to explain multi-seed evaluation and we have also highlighted that high performance gap persists even at 100 training epochs across mutiple seed, hence there is no single fixed evaluation criteria. Moreover, multi-seed relative ranking has been discussed in Section 5.5.2 for unseen-data with Table 6.
> 3. "Why does Table 1 stop at 2019 for macro-only search methods": Generally, there have been fewer macro-search works since the advent of DARTS. We have updated Table 2 (previously Table 1) with some more relevant works on macro-search. Further, we have also added a new Table 3 comparing our work to strong and recent works including DARTS-based methods.
>
> Thank you for your positive feedback and specially pointing us to forgotten important literature. We warmly welcome any further suggestions that may help us improve our work.

---

> ### Author Response · Authors · 2026-04-27
> **Updated Response to Reviewer QUK9**
>
> Regarding the following feedback:
>
>  "Fully document the relevant parameters of the 2025 NAS Unseen-Data Challenge and present the main result in a figure or table, including other competitors and baselines"
>
> The officially verifiable URL to the leaderboard has been added in the revised manuscript and Table 5 has been updated with the results of the baseline as well as competing solutions. Thank you for taking the time to review our work.

---

### Review · Reviewer_JLvy · 2026-03-10

**Summary Of Contributions:**

This paper studies neural architecture search (NAS) in the more realistic setting of generalizing to unseen datasets. To address this setting, the paper proposes a NAS framework that combines a compact macro search space over depth and width, full-dataset candidate evaluation, multi-seed assessment, and a dataset- and architecture-aware multi-fidelity search strategy designed for the NAS Unseen-Data Challenge. The paper’s main message is that low-fidelity, single-seed evaluation can produce unstable architecture rankings. Empirically, the method performs on the three challenge datasets and achieves good results under the imposed time budget.

Strengths
1. The paper addresses an important and practically relevant NAS setting, namely generalization to truly unseen datasets.
2. The overall framework is clear and easy to follow, with a coherent combination of compact search space design, full-dataset evaluation, multi-seed assessment, and multi-fidelity search.
3. The paper raises a valid concern that candidate rankings can vary across random seeds, which is an important issue for reliable NAS evaluation.

Weaknesses
1. The practical meaning of the 8-hour budget is somewhat unclear, since wall-clock comparisons may depend heavily on the available hardware rather than algorithmic efficiency alone. For example, methods run with faster GPUs or multiple GPUs may effectively be able to explore a larger search budget within the same nominal 8-hour constraint, which makes fairness harder to assess.
2. The search space and search procedure appear relatively simple, relying mainly on depth/width exploration and local search, which makes the methodological novelty feel somewhat limited. Since the search is confined to a small and structured space, it would be helpful to clarify whether simpler alternatives, such as binary search or other lightweight adaptive search strategies, could obtain comparable results more efficiently.
3. Some empirical results are not fully clarified, such as the apparent gap between the Conway validation behavior (70%) in Figure 4 and the final test accuracy (99%) in Table 2.
4. Given the paper’s emphasis on NAS for unseen datasets, the empirical case would be stronger if it included a more direct comparison between standard seen datasets such as ImageNet and the target unseen datasets, so that the distinction from conventional benchmark-oriented NAS is demonstrated more explicitly.
5. Although the related work covers several relevant baselines, the experimental section does not provide much direct comparison to them, which makes it harder to assess the proposed method against the broader NAS literature.

**Audience:**

No

**Audience Explanation:**

The current version does not yet provide sufficiently strong evidence or depth of analysis to make the findings compelling.

**Claims And Evidence:**

No

**Claims Explanation:**

Several core claims would benefit from stronger empirical support, clearer comparisons to prior baselines, and better clarification of some reported results.

**Requested Changes:**

I would like to see the weaknesses above addressed, in particular through clearer experimental justification of the main claims, stronger baseline comparisons, and clarification of the reported results.

---

> ### Author Response · Authors · 2026-04-19
> **Response to Reviewer JLvy**
>
> Thank you for your valuable feedback.
>
> 1. "The practical meaning of the 8-hour budget is somewhat unclear" : The proposed framework itself is flexible to run on any given time budget and larger search spaces can be explored with more resources. The 8-hour constraint is solely a requirement for the NAS Unseen-Data Challenge. Submission exceeding 24-hour runs on a single GPU across 3 datasets are penalized. We have clarified this in 5.1.2. Moreover, for our experiments on CIFAR-10 added in Section 4, we have run search without time constraints and have achieved competitive results.
> 2. "The search space and search procedure appear relatively simple": The search strategy comprises of two phases. Local search is used in Phase 1 only to collect an initial pool of promising candidates. Phase 2 is where multi-fidelity search algorithm iteratively eliminates candidates until the best model remains. We have clarified this in the introduction. Regarding binary search; if we assume there exists a training protocol which is equally good for all small/large candidates, then it is possible to apply binary search after an initial pool of candidates has been acquired in Phase 1. However, most of the compute is spent on multi-seed evaluation rather than number of candidates, therefore a significant speed-up is not expected.
> 3. "Some empirical results are not fully clarified": We believe we have been referred to figure 3 in the revised version. In the figure, 'Searched Model' accuracies are achieved during the search whereas 'Trained Model' accuracies are obtained after the final training. The numbers match with those reported in Table 5 (revised version).
> 4. "Given the paper’s emphasis on NAS for unseen datasets" : We have added Table 1 to detailing differences between NAS for seen versus unseen data.
> 5. "Although the related work covers several relevant baselines": We have added a new section (Section 4) containing comprehensive experiments to compare our work against broader NAS literature for both macro-search and DARTS-based search methods. Section 4.1 is about the contribution of macro-search whereas 4.2 discusses the applicability and effectiveness of the proposed search strategy on DARTS search space, singling out its contribution. We have also added Table 3 that contains comparisons against both macro- and recent micro-search methods and have achieved competitive results.
>
> We appreciate your valuable feedback and look forward to suggestions that can further improve our work.

---

### Review · Reviewer_Q819 · 2026-04-06

**Summary Of Contributions:**

In this manuscript, authors address a well-motivated limitation of current neural architecture search, i.e., poor generalization beyond canonical benchmarks (e.g., CIFAR/ImageNet). Authors argue that common NAS practices focused on large search spaces combined with low fidelity, which lead to unreliable candidate ranking and poor generalization.

To this end, they proposed a system-level NAS pipeline redesign, consists of three main components: 1) a compact macro-only search space restricted to depth and width, with dataset-adaptive constraints; 2) a high-fidelity evaluation protocol that trains each candidate on the full dataset and across multiple random seeds; and 3) a customized multi-fidelity local search procedure that allocates budget progressively in two phases. Empirically, authors claim that the proposed method is evaluated in the NAS Unseen-Data Challenge 2025 and achieved the top score under strict time constraints.

Overall, this manuscript is well-motivated and aiming to address a relevant limitation within current NAS pipeline. However, the contribution is primarily at the level of integration and design choices, rather than introducing fundamentally new algorithm, search spaces, or evaluation metrics.

**Audience:**

Yes

**Audience Explanation:**

This manuscript addresses an relatively important and underexplored problem, that is how NAS methods perform under distributional uncertainty and unseen datasets, rather than on well-established benchmark datasets. This could be interesting to researchers in NAS, AutoML, as well as efficient model design.

**Broader Impact Concerns:**

No major ethical concerns.

**Claims And Evidence:**

No

**Claims Explanation:**

Authors provide evidence that the overall pipeline is effective in the specific setting of the NAS Unseen-Data Challenge. However, several aspects of the empirical support is limited.

1. The main claim relies on reported challenge results, i.e., first place with score 12.19. However, the manuscript doesn't provide a verifiable reference, e.g., an official leaderboard, challenge report, or public ranking page. I was also unable to readily locate such resource. As a result, the reported results are currently not independently verifiable from the information provided, which weakens the strength of the empirical evidence.

2. The empirical evaluation doesn't sufficiently isolate the contributions of individual components. The proposed method combines multiple design choices, i.e., compact search space, multi-seed full-dataset evaluation, and a custom multi-fidelity search strategy, but doesn't provide controlled ablations.

3. While the manuscript argues that higher-fidelity evaluation improves ranking reliability, this claim isn't rigorously validated against alternative fidelity strategies under equal computation budgets.

4. The discussion about computational cost is incomplete, although authors acknowledge that full-dataset, multi-seed evaluation is expensive, it doesn't report key quantities such as the number of evaluated architectures, per-candidate cost, or total computation usage. This makes it difficult to asses efficiency or fairness to prior work.

**Requested Changes:**

1. **Provide verifiable evidence for challenge results. (Critical)** Include a link to an official leaderboard, challenge report, or any independently verifiable record of this competition. Without this, the primary empirical claim cannot be independently confirmed.

2. **Add controlled ablations to disentangle contributions. (Critical)** Under matched computational budgets, provide experiments to isolate:
    * the effect of the macro-only search space,
    * the impact of multi-seed evaluation,
    * the contribution of the proposed multi-fidelity strategy.

3. **Compare against prior methods/baselines under the same setting. (Critical)** Evaluate alternative methods using the same search space and time budget to determine whether the proposed search strategy provides meaningful and measurable gains.

4. **Quantify computational cost (Critical).** Full-dataset training is not new, but an old antique, which could lead to drastic computation budget. It's important to report:
   * number of evaluated architectures,
   * training cost per candidate, including number of seeds,
   * total compute usage, e.g., GPU hours,
   * trade-offs between fidelity and cost.

5. **Clarify ranking methodology under multi-seed evaluation. (Important)** Authors should specify whether candidates are ranked by mean, best, or another statistic across seeds, and justify this choice.

6. **Provide sensitivity analysis for key design choices. (Important)** Authors should analyze how performance depends on parameters such as search space bounds, number of seeds, etc.

7. **Clarify algorithmic details for reproducibility. (Important)** Authors should provide a more formal description of the two-phase search procedure, including candidate selection and budget allocation rules, etc.

---

> ### Author Response · Authors · 2026-04-19
> **Response to Reviewer Q819**
>
> We would like to thank you for your comprehensive and valuable guidance and feedback. Please find below our responses:
>
> 1. "Provide verifiable evidence for challenge results": We do have a verifiable certificate confirming first-place for the challenge. However, we are unable to share it due to double-blind review policy. We are in contact with challenge organizers to add a verifiable link containing challenge details and scores of other participants while keeping our identity hidden. However, we would like to request some time to add those details to paper.
> 2. "Add controlled ablations to disentangle contributions": We have added a new section (Section 4) containing comprehensive experiments to compare our work against broader NAS literature for both macro-search and DARTS-based search methods. Section 4.1 is about the contribution of macro-search whereas 4.2 discusses the applicability and effectiveness of the proposed search strategy on DARTS search space, singling out its contribution. Further section 3.2 has been updated to further explain multi-seed evaluation. Moreover, multi-seed relative ranking has been discussed in Section 5.5.2 for unseen-data with Table 6.
> 3. "Compare against prior methods/baselines under the same setting.": We have added Table 3 that contains comparisons against both macro- and recent micro-search methods and have achieved competitive results.
> 4. "Quantify computational cost": We have added computational requirements and evaluated architectures in section 4.1.3 and other respective sections as well. We have added Section 4.3 for ablation studies and discussed trade-offs between data fidelity, cost and impact on results.
> 5. "Clarify ranking methodology under multi-seed evaluation": We use the best-seed accuracy and have added reasons for doing so in Section 3.2.1.
> 6.  "Provide sensitivity analysis for key design choices.": We have added seed related cost effect in Ablation Studies section 4.3.2.
> 7. "Clarify algorithmic details for reproducibility": We have updated the algorithm to explain better and we would also release all codes, data, search and training logs for reproducibility.
>
> We have revised our manuscript and would once again thank you for your positive feedback as it has indeed helped us significantly improve our work and we look forward to any further suggestions you may have.

---

> > ### Author Response · Authors · 2026-04-27
> > **Updated Response to Reviewer Q819**
> >
> > Regarding the following feedback:
> >
> > 1. "Provide verifiable evidence for challenge results. (Critical) Include a link to an official leaderboard, challenge report, or any independently verifiable record of this competition. Without this, the primary empirical claim cannot be independently confirmed."
> >
> > The officially verifiable URL to the leaderboard has been added in the revised manuscript. Table 5 has also been updated with the results of the competing solutions. Thank you for taking the time to review our work.

---

### Decision · Action_Editor_XP8q · 2026-05-17

**Recommendation:** Accept as is

**Audience:**

Yes

**Audience Explanation:**

The clarifications on the results, including the evidence from the competition, make it clear that the proposed approach could benefit the NAS/ML community. It is a standard approach for AutoML. Given the standard benchmarks and comparisons, it is evident that the proposed approach would benefit a wide range of researchers.

**Claims And Evidence:**

Yes

**Claims Explanation:**

The original submission contained claims that were not fully supported. In particular, the claims regarding restricting the NAS search space, providing evidence about the NAS leaderboard and the discussion of related work were addressed in more detail in the updated version. Additionally, concerns regarding the experimental protocol were addressed in the revised version through the inclusion of more ablation studies and baseline comparisons, as well as additional information such as the computational cost. Overall, the updated version provides strong support for the claims made in the paper. It also effectively addresses all the criticism in the reviews.